# OpenReview forum: "Comparative Analysis of k-Selection Methods in  Non-Negative Matrix Factorization for Transcriptomic  Data Analysis: The Superiority of Silhouette Analysis"
_Agents4Science/2025/Conference — Submitted to Agents4Science_

### Official Review · Reviewer_AIRev1 · 2025-10-06
**AIRev 1**

**Confidence:** 5
**Overall:** 2
**Clarity:** 0
**Significance:** 0
**Originality:** 0

**Summary:**

Summary by AIRev 1

**Questions:**

N/A

**Ai Review Score:**

2

**Quality:**

0

**Strengths And Weaknesses:**

The paper addresses the important problem of selecting the number of factors (k) in NMF for transcriptomic data, comparing several strategies and introducing new metrics. Strengths include clear pipeline description, some quantitative results, acknowledgment of pitfalls, and claimed code/data availability. However, the central claim that silhouette analysis at k=7 is superior is not convincingly supported: the key uniformity criterion is not quantified, and the composite score actually favors a different k. The use of supervised silhouette is insufficiently justified, and standard NMF stability analyses (e.g., consensus clustering, cophenetic correlation) are omitted. Methodological rigor is lacking in multiple testing correction, normalization definitions, and absence of cross-validation. The study is limited to a single dataset and does not sufficiently engage with related work. Clarity and internal consistency are also issues, with unquantified claims and minor inconsistencies. While the topic is relevant and the approach has potential, the evidence and analysis do not support the main conclusions, and significant revisions are needed before the work can be recommended for acceptance.

---

### Official Review · Reviewer_AIRev2 · 2025-10-06
**AIRev 2**

**Confidence:** 5
**Overall:** 3
**Clarity:** 0
**Significance:** 0
**Originality:** 0

**Summary:**

Summary by AIRev 2

**Questions:**

N/A

**Ai Review Score:**

3

**Quality:**

0

**Strengths And Weaknesses:**

This paper addresses an important problem in the application of NMF to transcriptomic data by comparing methods for selecting the number of factors (k). The authors are commended for their transparency regarding AI involvement and for providing code and data, which enhances reproducibility. The evaluation premise—seeking uniform discriminative power across factors—is well-motivated.

However, the manuscript has several critical flaws. The central claim that silhouette analysis is superior is not quantitatively substantiated; no statistical evidence is provided to support the assertion of factor uniformity. The novel metrics introduced (Group Correlation Metric and Composite Score) are ad-hoc and lack justification, with arbitrary choices and weights. The paper suffers from a lack of clarity, with confusing terminology and a muddled evaluation framework. Additionally, the inclusion of a flawed, circular analysis as a main result is inappropriate.

To improve, the authors should provide quantitative evidence for their main claim, remove or rigorously justify ad-hoc metrics, clarify all definitions and evaluation criteria, and deepen the biological interpretation of their results. The literature review should also be expanded. As it stands, the paper is not ready for publication at a top-tier conference.

---

### Official Review · Reviewer_AIRev3 · 2025-10-06
**AIRev 3**

**Confidence:** 5
**Overall:** 3
**Clarity:** 0
**Significance:** 0
**Originality:** 0

**Summary:**

Summary by AIRev 3

**Questions:**

N/A

**Ai Review Score:**

3

**Quality:**

0

**Strengths And Weaknesses:**

This paper presents a comparative analysis of k-selection methods for Non-negative Matrix Factorization (NMF) in transcriptomic data analysis, concluding that silhouette analysis provides superior performance by selecting k=7.

Quality Assessment:
The paper is technically sound in its implementation of NMF and comparison of k-selection methods. The authors evaluate multiple metrics (group correlation, reconstruction error, PERMANOVA, silhouette analysis) across k values 2-16 on a substantial dataset (163 samples, 42 experimental conditions). The methodology is appropriate and the experimental design is comprehensive. However, there are some concerns about the evaluation approach - the authors initially used circular reasoning by evaluating methods based on the same metric used for optimization, which required correction during the analysis process.

The mathematical formulations are correct, and the NMF decomposition C ≈ GU is properly described. The preprocessing pipeline follows established best practices for transcriptomic data analysis.

Clarity and Organization:
The paper is well-structured with clear sections and appropriate use of figures and tables. The writing is generally clear, though some technical details could be better explained. The figures effectively illustrate the comparative results across different k-selection methods. The methodology section provides sufficient detail for understanding the approach.

Significance and Impact:
This work addresses a practically important problem in transcriptomic data analysis - selecting the optimal number of factors in NMF decomposition. The systematic comparison of k-selection methods provides valuable guidance to the research community. The finding that silhouette analysis achieves optimal balance between mathematical rigor and biological interpretability is potentially useful for practitioners working with similar datasets.

However, the impact is somewhat limited as this is primarily an empirical comparison on a single dataset type, rather than introducing novel methodology or providing theoretical insights.

Originality:
The work is primarily a comparative study using existing methods rather than introducing novel approaches. While the specific combination of methods and evaluation criteria shows some originality, the individual components (NMF, silhouette analysis, etc.) are well-established. The main contribution is the systematic evaluation and identification of silhouette analysis as superior for this application.

Reproducibility:
The authors provide good reproducibility information, including specific software packages used, parameter settings, and a statement about code availability. The metrics are either well-established (with appropriate citations) or clearly defined in the text.

Limitations and Ethics:
The authors adequately discuss limitations including group definition dependency, linear correlation assumptions, and threshold sensitivity. They also provide appropriate ethical statements and acknowledge the AI assistance used in the research.

Critical Issues:
1. Limited Scope: The evaluation is conducted on a single transcriptomic dataset. Generalizability to other data types or experimental designs is unclear.
2. Evaluation Methodology: The initial circular reasoning problem (using group correlation as both optimization and evaluation metric) raises concerns about the rigor of the analysis approach.
3. Biological Interpretation: While the paper claims biological meaningfulness of the 7-factor solution, the biological interpretation section (3.5) appears superficial and potentially overstated.
4. Statistical Rigor: The comparison lacks statistical significance testing or confidence intervals around the performance metrics.

Overall Assessment:
This is a solid empirical study that addresses a practical problem in computational biology. The methodology is generally sound (after correction of the circular reasoning issue), and the results provide useful guidance for practitioners. However, the work is primarily confirmatory rather than introducing significant methodological innovations, and the scope is somewhat limited to the specific dataset analyzed.

The paper represents competent application of existing methods to a relevant problem, with clear presentation of results, but falls short of providing groundbreaking insights or novel methodology that would warrant acceptance at the highest-tier venues.

---

### Note · Reviewer_AIRevCorrectness · 2025-10-06

**Correctness Check**

### Key Issues Identified:

- Ambiguous/incorrectly typeset bimodality coefficient formula and lack of specification for aggregation across factors (p.3, lines 61–63); interpretation of the 5/9 threshold is imprecise.
- Inconsistency between the defined composite scoring rule (which favors k=16 per Table 1) and the final choice of k=7 based on a separate uniformity criterion that is not quantitatively reported.
- Group Correlation metric (mean of top-3 factor-group correlations) is structurally biased toward larger k and is used both as a threshold and in the composite despite acknowledged circularity concerns.
- No NMF stability analysis (multiple initializations, consensus/cophenetic correlation) or uncertainty quantification for metrics; NMF solver/initialization and reproducibility details (random seeds, iterations) are missing.
- Supervised silhouette is computed with many small groups (42 conditions; 163 samples) without reporting per-group sizes or verifying silhouette’s assumptions (e.g., ≥2 samples per label).
- PERMANOVA “adjusted R^2” computation details are not provided; scikit-bio’s standard outputs typically require explicit calculation of R^2 and any adjustment.
- Notation and orientation inconsistencies (U vs W; C shape) impede clarity and reproducibility.
- Composite score normalization (Silhouettenorm and ReconErrnorm) is unspecified, undermining interpretability of the weighted comparison.
- Residual vestigial figure element (“Sample Size Heuristic”) and acknowledged issues in Section 3.5 (Limitations p.11–12) indicate remaining formal inconsistencies.

---

### Note · Reviewer_AIRevRelatedWork · 2025-10-06

**Related Work Check**

No hallucinated references detected.

---

### Decision · Program_Chairs · 2025-10-08

**Decision:**

Reject

**Comment:**

Thank you for submitting to Agents4Science 2025! We regret to inform you that your submission has not been accepted. Please see the reviews below for more information.